# Predictive Value of Annenxin A1 for Disease Severity and Prognosis in Patients with Community-Acquired Pneumonia

**DOI:** 10.3390/diagnostics13030396

**Published:** 2023-01-21

**Authors:** Minghao Gu, Xiudi Han, Xuedong Liu, Fengxiang Sui, Quansan Zhang, Shengqi Pan

**Affiliations:** 1Qingdao Municipal Hospital Affiliated to Qingdao University, Qingdao 266011, China; 2Medical School of Qingdao University, Qingdao 266071, China

**Keywords:** annenxin A1, community-acquired pneumonia, severity, mortality

## Abstract

This prospective, single-center study evaluated the clinical utility of annenxin (Anx)A1 level as a biomarker for determining the severity of illness and predicting the risk of death in hospitalized patients with community-acquired pneumonia (CAP). A total of 105 patients (53 with severe [S]CAP, 52 with non-SCAP) were enrolled from December 2020 to June 2021. Demographic and clinical data were recorded. Serum AnxA1 concentration on days one and six after admission was measured by enzyme-linked immunosorbent assay. AnxA1 level at admission was significantly higher in SCAP patients than in non-SCAP patients (*p* < 0.001) irrespective of CAP etiology and was positively correlated with Pneumonia Severity Index and Confusion, Uremia, Respiratory Rate, Blood Pressure, and Age ≥ 65 Years score. AnxA1 level was significantly lower on day six after treatment than on day one (*p* = 0.01). Disease severity was significantly higher in patents with AnxA1 level ≥254.13 ng/mL than in those with a level <254.13 ng/mL (*p* < 0.001). Kaplan–Meier analysis of 30-day mortality showed that AnxA1 level ≤670.84 ng/mL was associated with a significantly higher survival rate than a level >670.84 ng/mL. These results indicate that AnxA1 is a useful biomarker for early diagnosis and prognostic assessment of CAP.

## 1. Introduction

Community-acquired pneumonia (CAP) poses a major threat to public health as an infectious disease with high morbidity and mortality [1]. Accurate assessment of CAP severity and prognosis is critical for diagnosis and treatment. Comprehensive scoring systems, such as Confusion, Uremia, Respiratory Rate, Blood Pressure, and Age ≥ 65 Years (CURB-65) score and Pneumonia Severity Index (PSI) as well as highly specific and sensitive biomarkers that can help to determine pneumonia severity are currently recommended and widely used for this purpose [2,3]. However, CURB-65 only evaluates a few vital signs and laboratory indices and may have low specificity in certain populations such as the elderly [4]. Meanwhile, PSI score is based on age, underlying diseases, and laboratory results that are difficult to measure. Biomarkers can be useful for diagnosis and for monitoring the response to anti-infective therapy.

Annexin (Anx)A1 (also known as lipocortin 1) is a 37-kDa phospholipid-binding protein that is expressed in many tissues and cell types, including leukocytes, monocytes, and epithelial cells. AnxA1 participates in the regulation of endo- and exocytosis, signal transduction, cellular metabolism, and cytoskeletal rearrangement, all of which are important for cell proliferation, differentiation, migration, survival, and repair. AnxA1 was shown to be involved in the antithrombotic response and response to viral infection [5,6,7,8,9], and its role in suppressing inflammation was demonstrated in AnxA1 knockout mice (AnxA1^−/−^) based on glucocorticoid-induced activation of innate immune cells [10]. Dysregulation of AnxA1 expression has been reported in patients with novel coronavirus disease 2019 (COVID-19) [11,12]. Multiplatform omics analysis of serial plasma and urine samples collected from patients during the course of COVID-19 illness identified AnxA1 as a potential therapeutic target: Serum AnxA1 level was significantly lower in patients with severe or critical illness compared with the control and moderate illness groups [11]. AnxA1 was also shown to be a diagnostic biomarker for COVID-19 pneumonia and predicted the need for intensive care unit (ICU) treatment in these patients with 69.8% sensitivity and 58.1% specificity [12]. However, it is unclear whether AnxA1 expression has predictive value for nonviral pneumonia.

We hypothesized that serum AnxA1 level may be correlated with disease severity and mortality in CAP. The aim of this study was to clarify the role of AnxA1 in CAP and validate the utility of AnxA1 as an index of illness severity and risk of death in severe (S)CAP.

## 2. Materials and Methods

### 2.1. Study Design

This study was conducted at Qingdao Municipal Hospital, Qingdao, Shandong Province, China, from December 2020 to June 2021. Ethics approval was obtained from the Ethics Committee of Qingdao Municipal Hospital (no. 2020CXJJ001-052). Patients were informed in detail about the study and were requested to complete the written consent forms before participating. All patients diagnosed with CAP in this study were recruited from the ICU, emergency room, or respiratory medicine department. Patients had to meet criterion A, criterion C, and any one the conditions of criterion B [13] to be included in the study. (A) Onset in the community setting. (B) Relevant clinical manifestations of pneumonia, including (1) new onset of cough or expectoration or aggravation of existing symptoms of respiratory tract diseases, with or without purulent sputum, chest pain, dyspnea, or hemoptysis; (2) fever; (3) signs of pulmonary consolidation and/or moist rales; and (4) peripheral white blood cell (WBC) count >10 × 10^9^/L or <4 × 10^9^/L, with or without a left shift. (C) Chest radiograph showing new patchy infiltrates, lobar or segmental consolidation, ground-glass opacities, or interstitial changes, with or without pleural effusion. Exclusion criteria were as follows: Tuberculosis, pulmonary tumor, noninfectious interstitial lung disease, pulmonary edema, atelectasis, pulmonary embolism, pulmonary eosinophilia, or pulmonary vasculitis. Pregnant women and patients who were immunocompromised, including those with a history of taking glucocorticoid for more than 1 month, history of immunosuppressive therapy, human immunodeficiency virus infection, solid tumor or hematologic malignancy, or recent hospitalization (<30 days) were also excluded.

### 2.2. Data Collection

We set up a case report form to collect data including age, sex, height, weight and other demographic information; clinical symptoms and signs; and treatment before admission and during hospitalization including antiviral drugs, hormones, and antibiotics. The following laboratory parameters were measured within 24 h of admission: routine blood parameters, biochemical indices, C-reactive protein (CRP), procalcitonin (PCT), etc. Etiologic examination included blood, sputum, and bronchoalveolar lavage fluid; chest radiography and/or chest computed tomography; comorbidities; length of hospital stay; and outcome. CURB-65 score and PSI were determined for each patient. Outcomes were assessed at 30 days after study enrollment through structured telephone interviews.

### 2.3. Measurement of Serum AnxA1 Level

Peripheral venous blood was collected in a sterile procoagulation tube on days 1 and 6 of hospitalization and centrifuged within 3 h at 3000 rpm for >10 min. The serum was separated and stored at −80°C until analysis. AnxA1 level in serum samples was measured with quantitative enzyme-linked immunosorbent assay kits (Boster Bio, Pleasanton, CA, USA) according to the manufacturer’s instructions. During the assay, interference elimination reagents were added to the dilutions to eliminate interference from rheumatoid factors, hama, etc., and negative controls were made, using dilutions and normal serum as negative controls, respectively. All data are presented after subtracting the negative control, and the error caused by the interfering items has been removed.

### 2.4. Statistical Analysis

Categorical variables are reported as numbers or percentages and were assessed with Fisher’s exact test or the chi-squared test. The distribution of continuous variables was evaluated with the Kolmogorov–Smirnov test, and the variables are expressed as mean ± SD when the data showed a normal distribution, with a post hoc Tukey honestly significant difference test or Student’s *t* test used to evaluate the significance of intergroup differences in the data. If the normality assumption was violated, data are reported as median and interquartile range (IQR) and were analyzed with the Kruskal–Wallis H test or Mann–Whitney U test. Receiver operating characteristic (ROC) curve analyses were performed using the optimal threshold determined by the Youden index, and the area under the curve (AUC) and 95% confidence interval (CI) were calculated to assess their predictive value. The 2-tailed Spearman rho coefficient was calculated, and an adjusted Bonferroni correction was applied to evaluate the strength and direction of the linear relationship between AnxA1 level and clinical indicators. The Kaplan–Meier method was used to generate a 30-day survival curve, and the log–rank test was used to compare survival rates. Statistical tests were 2-sided, with significance determined at *p* < 0.05. SPSS v23.0 (IBM, Armonk, NY, USA) and Prism v8.0.1 (GraphPad Software, San Diego, CA, USA) were used to conduct all statistical tests.

## 3. Results

### 3.1. Patient Characteristics

The study population comprised 105 patients with CAP who were divided into non-SCAP (n = 52) and SCAP (n = 53) groups. The median age of the participants was 73.0 years (IQR, 59.0–86.0 years). A total of 14 SCAP patients died during hospitalization. The demographic and clinical characteristics of the participants are presented in Table 1. There were significant differences between the SCAP and non-SCAP groups in terms of sex ratio and age range (*p* < 0.05 for both groups), which affected the comparability between the 2 groups: patients in the SCAP group were older whereas the non-SCAP group had more males. In laboratory tests, indicators of an inflammatory response such as WBC count, neutrophil (NEU) count, lymphocyte (LYM) count, neutrophil-to-lymphocyte ratio (NLR), and PCT level differed significantly between SCAP and non-SCAP patients (all *p* < 0.05). Chest radiographs revealed that 84.9% of patients with SCAP showed multipolar infiltrates and 50.9% showed pleural effusion; these proportions were significantly higher than in the non-SCAP group (46.2% and 11.5%, respectively; *p* < 0.001 for both comparisons).

The CURB-65 score and PSI in the non-SCAP group were 0 (0–1) and 65.83 ± 26.26, respectively, which were significantly lower than those in the SCAP group (3 [2,3] and 143.94 ± 32.50, respectively; both *p* < 0.001). Thirteen patients (12.4%) died within 30 days, and one patient (0.9%) died on day 58; 86 (81.9%) patients were discharged within 30 days with improved health conditions (median: 9.00 days; range: 7.75–13.00 days), and five (4.8%) were discharged after more than 30 days of hospitalization (median: 37 days; range: 34.5–66.5 days).

### 3.2. AnxA1 Level and CAP Etiology

Serum AnxA1 level at admission was significantly lower in the non-SCAP group than in the SCAP group (240.76 [202.42–378.60]) vs. 441.91 [282.25–780.33] ng/mL, *p* < 0.001; Figure 1a). Meanwhile, AnxA1 level was significantly higher in nonsurvivors than in survivors (758.02 [372.39–976.16]) vs. 289.32 [218.09–465.81] ng/mL, *p* < 0.001; Figure 1b). A total of 91 CAP patients were discharged with improved health conditions after treatment; of the 31 patients for whom samples were obtained on day six of hospitalization, 23 showed decreased serum AnxA1 concentrations. Among survivors, mean AnxA1 concentration was lower on day six (median: 289.32 ng/mL; range: 218.09–465.81 ng/mL) than on day one (median: 231.37 ng/mL; range: 199.47–251.83 ng/mL) (*p* = 0.01; Figure 1c).

The overall pathogen detection rate was 21.0%, and there was more etiologic evidence in the SCAP group (44.0%) than in the non-SCAP group (7.7%). The most common pathogen was *Acinetobacter baumannii*, followed by *Pseudomonas aeruginosa* and *Klebsiella pneumoniae*. Fungi were detected in four cases and one patient was infected with atypical pathogens. One patient was infected with both bacteria and fungi. No viral infection was observed in the study population. There was no significant association between AnxA1 level and specific etiologic agents (*p* = 0.78; Figure 1d).

### 3.3. Case-Control Matching

To exclude the influence of age differences on AnxA1 level, cases and controls were matched 1:1 (n = 26 per group) with a five-year age difference as caliper. After matching, the gender difference between the two groups was also eliminated. The results showed that there was still a significant difference in AnxA1 level between the two groups (*p* < 0.001; Table 2).

### 3.4. Correlation between AnxA1 Level and CAP Severity

Serum AnxA1 level was significantly higher in CAP patients with high CURB-65 scores (3–5) than in those with low CURB-65 scores (0–2) (*p* < 0.001; Figure 2a); it was also higher in patients classified as high risk based on PSI (class IV or V) than in those classified as low risk (I–III) (*p* < 0.001; Figure 2b). A correlation analysis showed that serum AnxA1 level at admission was positively correlated with CURB-65 score and PSI (r = 0.453; r = 0.429, respectively; both *p* < 0.001). Additionally, AnxA1 level at admission was positively correlated with WBC count (r = 0.354, *p* = 0.010), NEU count (r = 0.378, *p* = 0.006), NLR (r = 0.352, *p* = 0.011), PCT level (r = 0.354, *p* = 0.011), IL-6 level (r = 0.402, *p* = 0.003), IL-8 level (r = 0.446, *p* < 0.001), and IL-10 level (r = 0.352, *p* < 0.001) (Figure 3).

### 3.5. Predictive Value of AnxA1 Level for SCAP in Patients with CAP

We investigated risk factors for predicting SCAP by binary multivariate logistic regression analysis (Table 3). PSI (hazard ratio [HR] = 1.111, 95% CI: 1.047–1.179; *p* < 0.001) and CURB-65 score (HR = 10.883, 95% CI: 3.231–36.659; *p* < 0.001) were independent predictors of SCAP after adjusting for laboratory indices such as WBC, NEU and LYM counts, NLR, and IL-6, IL-8 and IL-10 levels.

The AUC for AnxA1 level in the ROC curve analysis was 0.768 (*p* = 0.001), and the optimal threshold AnxA1 concentration for predicting SCAP was 254.13 μg/mL (84.6% sensitivity, 64.0% specificity). The AUCs of CURB-65 score and PSI were 0.920 (0.850–0.990) and 0.955 (0.906–1.000), respectively; and the AUCs of IL-6; IL-8 and IL-10 were 0.675 (0.526–0.825), 0.823 (0.708–0.938) and 0.512 (0.351–0.674), respectively. Thus, CURB-65 score and PSI had greater predictive power than AnxA1 level for SCAP. Combining AnxA1 concentration with CURB-65 score improved the overall accuracy of prediction from 0.920 to 0.938, although no improvement was observed by combining AnxA1 with PSI (Figure 4 and Table 4).

### 3.6. AnxA1 Level in Predicting Prognosis in Patients with SCAP (30-Day Mortality)

We investigated whether AnxA1 level could be used to predict 30-day mortality in patients with SCAP by Cox proportional hazard regression analysis (Table 5 and Table 6). The AUC for the ROC curve based on AnxA1 level was 0.809 (*p* = 0.004), and the optimal threshold AnxA1 concentration for a prognosis of death was 670.84 μg/mL (66.7% sensitivity, 93.0% specificity). This AUC was slightly less than that of PSI (0.933), but higher than that of CURB-65 (0.685) (Figure 5). Cox proportional regression analysis of the predictive value for SCAP patient survival revealed that IL-6 level (HR = 1.013, 95% CI: 11.004–1.022; *p* = 0.006) and PSI score (HR = 1.034, 95% CI: 1.014–1.054; *p* = 0.001) were independent predictors of 30-day mortality after adjusting for IL-8, IL-10, AnxA1 level, and CURB-65 score (Table 6).

Kaplan–Meier survival curves were used to assess the relationship between serum AnxA1 level and 30-day mortality in patients with CAP (Figure 6). The optimal cutoff value of AnxA1 level for predicting 30-day mortality was 670.84 ng/mL. Patients with CAP were divided into two groups according to the optimal cutoff value (>670.84 [high] and ≤670.84 [low] ng/mL) calculated by ROC curve analysis. There was a statistically significant difference in mortality rate between low and high AnxA1 groups (log-rank χ^2^ = 6.890, *p* < 0.0001). The risk of death was 5.770 (1.336–24.920) times greater in the high AnxA1 group than in the low AnxA1 group.

## 4. Discussion

To our knowledge, this is the first study investigating the utility of AnxA1 level as a biomarker in nonviral pneumonia. The major findings were as follows: (1) Serum AnxA1 level was significantly higher in SCAP patients than in non-SCAP patients, particularly in nonsurvivors, but was unrelated to CAP etiology. (2) A higher AnxA1 level was positively correlated with PSI and CURB-65 score. (3) The threshold concentration of AnxA1 for distinguishing SCAP from non-SCAP was 254.13 ng/mL, with 84.6% sensitivity and 64.0% specificity; and the ROC curve analysis indicated that combining AnxA1 with CURB-65 score increased the prediction of SCAP.

The annexin family comprises two evolutionarily conserved and structurally related Ca^2+^ and phospholipid-binding proteins. Annexins interact with a variety of binding partners via their N terminus, which partly explains their functional diversity [14]. AnxA1 is a glucocorticoid-dependent protein that is present in many tissues including lungs, bone marrow, and intestine at concentrations <50 ng/mL. Upon glucocorticoid stimulation, AnxA1 is secreted by cells into the extracellular space to transduce the anti-inflammatory effects of glucocorticoid [15,16,17]. Thus, AnxA1 plays an important role in chronic inflammatory disorders such as arthritis [18] and chronic obstructive pulmonary disease [19]. AnxA1 level is elevated in the plasma of individuals with type 1 diabetes independent of renal function impairment [20]. High AnxA1 expression was shown to be associated with increased serosal invasion and peritoneal metastasis. Gastric cancer patients with high AnxA1 expression in tumors had worse overall survival compared with patients with low or no AnxA1 expression in tumors [21]. In the present study, SCAP was associated with an elevated serum AnxA1 concentration compared with non-SCAP, especially in nonsurvivors, suggesting that AnxA1 is a useful biomarker for the pathologic status of SCAP.

When stratified according to PSI and CURB-65 score, patients with greater illness severity score had a higher AnxA1 level at admission than those with a low severity score. We also found a strong correlation between AnxA1 level and various clinical and laboratory indices including WBC and NEU counts; NLR; and PCT, IL-6, IL-8, and IL-10 levels. IL-6 [22], IL-8 [23], and IL-10 [24] are nonspecific inflammatory cytokines that reflect the degree of inflammation and have prognostic value in certain inflammatory diseases. NLR is a convenient, readily determined biomarker that is used to assess mortality risk and predict prognosis in patients with tumors [25] and inflammation [26]. The levels of these factors were shown to be positively correlated with the severity of infection; our results demonstrate that serum AnxA1 level can be used to evaluate the severity of CAP.

The combination of AnxA1 level and CURB-65 score weakly improved the predictive accuracy of CURB-65 score for SCAP (i.e., from 0.920 to 0.938), but not when combined with PSI, likely due to the high accuracy of PSI-based diagnosis in this cohort. However, the predictive utility of AnxA1 for SCAP was significant, and AnxA1 was superior to other single laboratory indices, such as CRP, PCT, IL-6 and IL-10. The prognostic value of AnxA1 for COVID-19 was demonstrated by the observation that serum AnxA1 level was significantly lower in patients with severe or critical disease than in those with moderate disease or healthy control subjects, which is contrary to our findings; in the ROC curve analysis, the AUC for serum AnxA1 level in patients who needed ICU treatment was 0.701 (95% CI = 0.582–0.819; *p* = 0.003) with 69.8% sensitivity and 58.1% specificity at a cutoff of 17.2 ng/mL [12]. The fact that we observed the opposite result may be attributed to a decrease in endogenous glucocorticoids due to intensification of COVID-19 and sepsis. Thus, pathogen factors should be considered when assessing the prognosis and severity of CAP based on measurement of AnxA1 concentration.

The Kaplan–Meier survival analysis showed that CAP patients with serum AnxA1 level <670.84 ng/mL at admission had a significantly longer survival time compared with patients with a level ≥670.84 ng/mL. This implies that serum AnxA1 level at admission can serve as a biomarker for determining disease severity in patients with CAP, and can guide clinical management to improve patient survival.

In patients with COVID-19, AnxA1 level was shown to be inversely associated with the degree of lung involvement [27]. This is consistent with the results of a metabolomic analysis of severe COVID-19 patients, which found that AnxA1 plays an important role in infection with severe acute respiratory syndrome coronavirus (SARS-CoV-2)—the causative agent of COVID-19—and is a potential therapeutic target [11]. Ac2-26, a mimetic peptide of full-length AnxA1 protein, shares the anti-inflammatory effects of glucocorticoids [28] and reduced cytokine storm syndrome especially in patients with severe COVID-19 illness, suggesting that it is a promising treatment for patients with severe respiratory symptoms and multiple organ involvement [29]. A meta-analysis of studies evaluating treatments for CAP showed that systemic corticosteroid therapy reduced mortality rate by 3%, the need for mechanical ventilation by 5%, and duration of hospital stay by 1 day [30]. Taken together, the existing evidence and our results suggest that an increase in serum AnxA1 concentration predicts the progression of CAP to SCAP and the need to initiate glucocorticoids, although additional studies are needed to confirm this possibility.

The AnxA1/N-formyl peptide receptor (FPR)2 axis regulates the anti-inflammatory response through multiple signaling pathways including mitogen-activated protein kinase (MAPK; eg, extracellular signal-regulated kinase [ERK]1/2 and p38 MAPK), protein kinase B (Akt), and c-Jun N-terminal kinase as well as through increased intracellular Ca^2+^ concentration [31]. Additionally, the dephosphorylated form AnxA1 inhibits phospholipase (PL)A2, which plays a key role in the Lands cycle, a deacylation/acylation reaction involving glycerophospholipids such as phosphatidylcholine and phosphatidylethanolamine [32]. We speculate that AnxA1 level is associated with lipid metabolism in patients with CAP.

Our study had certain limitations. First, serum AnxA1 level was only measured at the time of and on day 6 after admission; we were unable to obtain serial AnxA1 measurements in all patients, which is not always feasible in a real-world setting. Second, there was no healthy control cohort to support our results; a larger, multicenter study is therefore needed to confirm our findings.

## 5. Conclusions

In conclusion, AnxA1 level at admission was significantly higher in patients with CAP with higher severity scores, and was unrelated to CAP etiology; moreover, an elevated AnxA1 level strongly predicted SCAP in patients with CAP. Thus, assessment of serum AnxA1 concentration at hospital admission may provide valuable prognostic information that can guide clinical management and thereby prevent the progression of CAP to more severe illness.

## Figures and Tables

**Figure 1 diagnostics-13-00396-f001:**
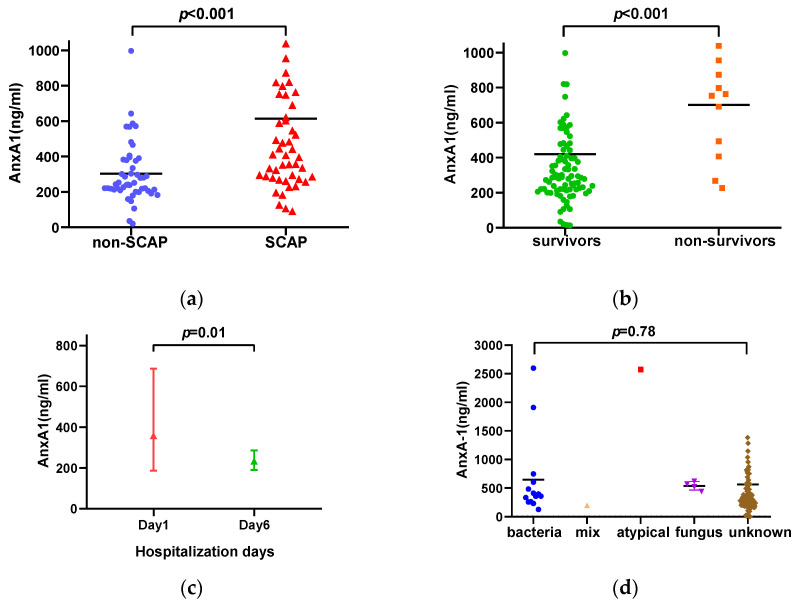
Serum AnxA1 level in patients with CAP. (**a**,**b**) AnxA1 level in patients with non-SCAP (blue circles) and SCAP (red triangles) (**a**) and in survivors (green circles) and non-survivors (orange squares) (**b**). (**c**) Serial change in serum AnxA1 concentration after treatment. (**d**) Comparison of AnxA1 level in patients with different CAP etiologies including bacteria, atypical pathogen (including *Mycoplasma pneumoniae*, *Chlamydia pneumoniae*, and *Legionella pneumophila*), mixed pathogen, and unknown pathogen. *p* = 0.78 for intergroup comparison. Each point represents the median value.

**Figure 2 diagnostics-13-00396-f002:**
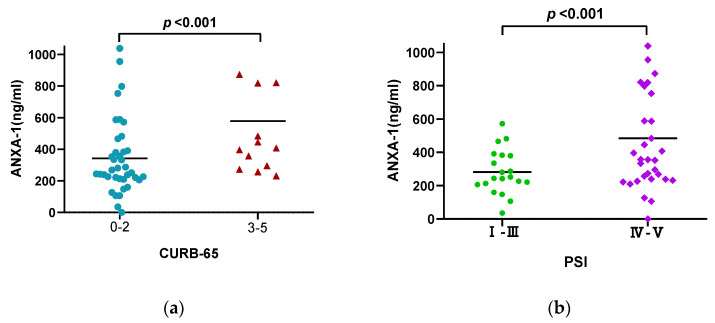
Distribution of AnxA1 by CURB-65 score and PSI class. (**a**) Statistically significant differences in AnxA1 level were observed among CAP patients according to the CURB-65 score. (**b**) AnxA1 level distinguished patients with high-risk PSI (classes IV and V). Each point represents the median value.

**Figure 3 diagnostics-13-00396-f003:**
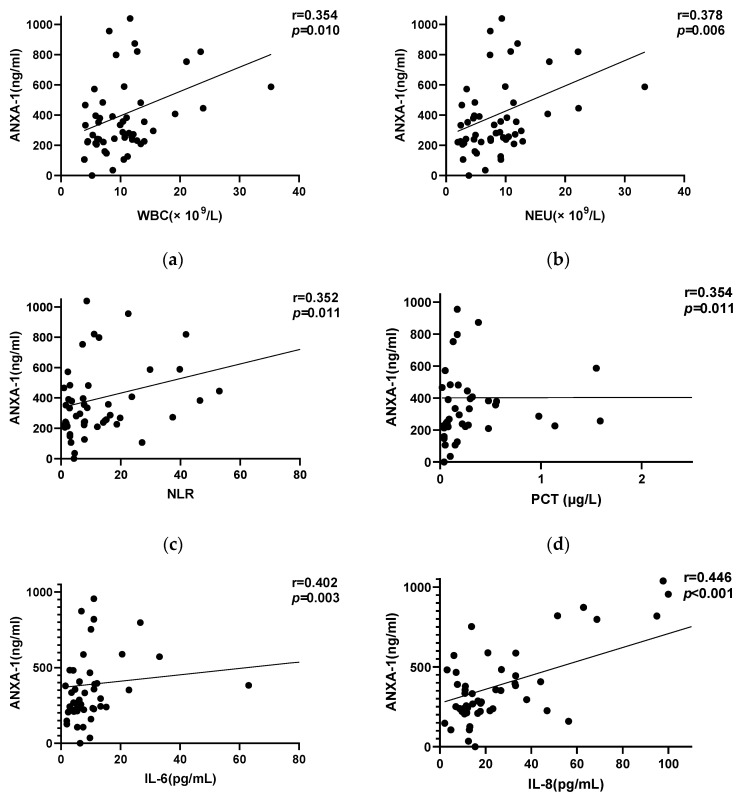
Correlation between AnxA1 level with multiple clinical indicators in 52 patients with CAP. (**a**–**i**) AnxA1 level showed a significant positive correlation with WBC (**a**), NEU (**b**), NLR (**c**), PCT (**d**), IL-6 (**e**), IL-8 (**f**), IL-10 (**g**), CURB-65 (**h**), and PSI score (**i**).

**Figure 4 diagnostics-13-00396-f004:**
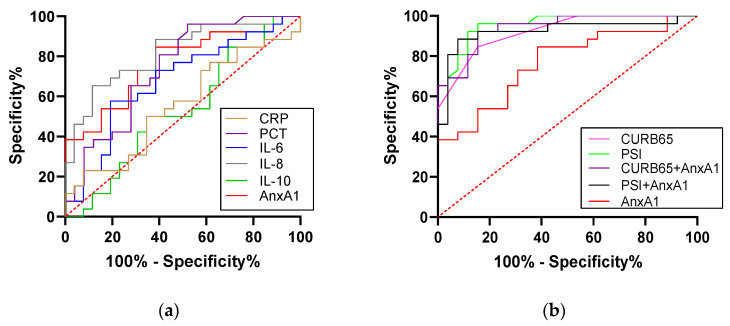
ROC curve analysis of the discriminatory capacity of AnxA1 level in CAP patient subsets. (**a**,**b**) ROC curve analysis of the capacity of AnxA1 and clinical indicators (**a**) and AnxA1 in combination with PSI or CURB-65 score (**b**) to discriminate between patients with SCAP and those with CAP.

**Figure 5 diagnostics-13-00396-f005:**
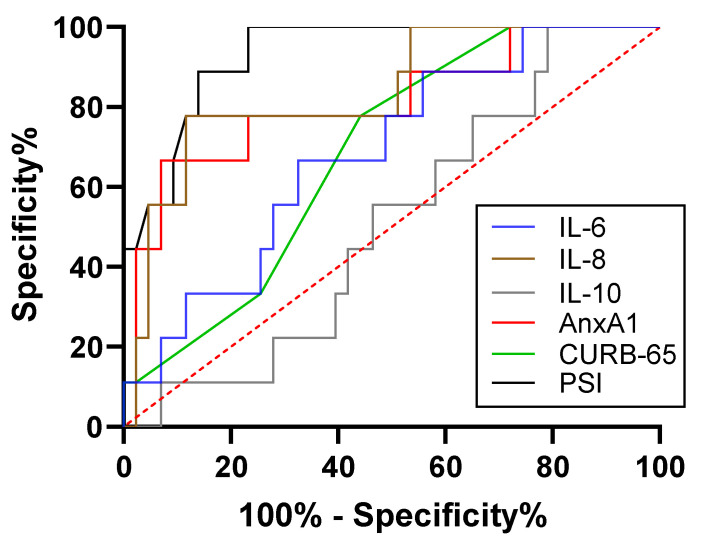
ROC curve analysis of the discriminatory capacity of AnxA1, IL-6, IL-8 and IL-10 levels; PSI and CURb-65 score for predicting 30-day mortality in patients with CAP.

**Figure 6 diagnostics-13-00396-f006:**
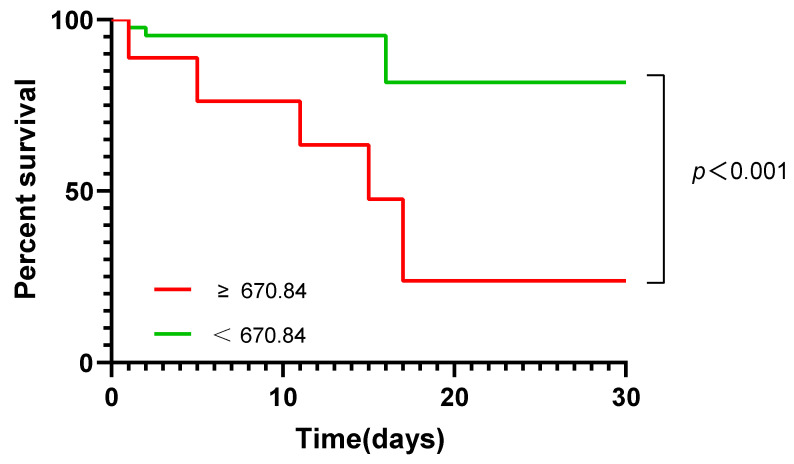
Kaplan–Meier analysis of 30-day mortality in CAP patients. The analysis was stratified by AnxA1 level. The AnxA1 cutoff (670.84 ng/mL) was an optimal retrospectively calculated value.

**Table 1 diagnostics-13-00396-t001:** Demographic and clinical characteristics of the CAP patients in this study.

Characteristic	Non-SCAP (N = 52)	SCAP (N = 53)	*p* Value
Sex, male (%)	40 (76.9)	27 (50.9)	0.006
Age, years	57.48 ± 18.93	81.13 ± 10.84	<0.001
Underlying disease(s)			
Chronic heart failure	1 (1.9)	15 (28.3)	<0.001
Coronary heart disease	4 (7.7)	32 (60.4)	<0.001
Diabetes mellitus	4 (7.7)	22 (41.5)	<0.001
Hypertension	20 (38.5)	33 (62.3)	0.015
Cerebrovascular disease	5 (9.6)	28 (52.8)	<0.001
Chronic liver disease	0	2 (3.8)	0.495
Chronic renal disease	1 (1.9)	8 (15.1)	0.031
Chronic obstructive pulmonary disease	0	7 (13.2)	0.013
Physical examination			
T_max_, °C	37.10 ± 0.86	37.1 (36.45–37.80)	0.468
Respiratory frequency, breaths/min	20.40 ± 2.01	21 (19.50–27.00)	0.051
Heart rate, beats/min	83.50 ± 9.87	98.98 ± 18.33	<0.001
Confusion	2 (3.8)	40 (75.5)	<0.001
SBP < 90 mm Hg or DBP ≤ 60 mmHg	0	4 (7.5)	0.118
Laboratory results			
WBC (×10^9^/L)	7.91 (5.85–10.65)	11.07 (8.73–14.75)	<0.001
NEU (×10^9^/L)	5.15 (3.43–8.91)	9.22 (7.41–12.76)	<0.001
LYM (×10^9^/L)	1.35 (0.89–1.72)	0.71(0.41–1.26)	<0.001
NLR	4.26 (2.09–9.07)	14.74 (6.92–25.41)	<0.001
CRP, mg/L	36.55 (11.44–106.06)	53.72 (24.69–112.78)	0.127
PCT, μg/L	0.10 (0.04–0.31)	0.38 (0.17–3.14)	<0.001
Interleukin-6, pg/mL	4.12 (3.15–7.25)	10.66 (5.46–16.37)	<0.001
Interleukin-8, pg/mL	11.46(9.1–15.52)	27.66 (14.59–51.09)	<0.001
Interleukin-10, pg/mL	9.25 (7.25–14.22)	12.46 (9.34–20.6)	0.007
AnxA1, ng/mL	240.76 (202.42–378.60)	441.91 (282.25–780.33)	<0.001
Chest X-ray			
Multipolar infiltrates	24 (46.2)	45 (84.9)	<0.001
Pleural effusion	6 (11.5)	27 (50.9)	<0.001
Pathogen established			
Bacteria	3 (5.8)	13 (24.5)	
Fungus	1 (1.9)	3 (5.7)	
Virus	0	0	
Atypical pathogen	0	1 (1.9)	
Mixed pathogen	0	1 (1.9)	
Unknown	48 (92.3)	35 (66.0)	
CURB-65 score			
Score points	0 (0–1)	3 (2–3)	<0.001
0	33 (63.5)	0	
1	14 (26.9)	5 (9.4)	
2	4 (7.7)	16 (30.2)	
3	1 (1.9)	26 (49.1)	
4	0	5 (9.4)	
5	0	1 (1.9)	
PSI			
Score points	65.83 ± 26.26	143.94 ± 32.50	<0.001
≤70	30 (57.7)	0	
71–90	11 (21.2)	2 (3.8)	
91–130	11 (21.2)	17 (32.1)	
>130	0	34 (64.2)	
30-Day mortality	0	13 (24.5)	<0.001

Variables are expressed as numbers (percentages); mean ± SD for continuous variables conforming to a normal distribution; and median (interquartile range) for continuous nonparametric data. Abbreviations: CAP, community-acquired pneumonia; CRP, C-reactive protein; CURB-65, Confusion, Uremia, Respiratory Rate, Blood Pressure, and Age ≥65 Years; DBP, diastolic blood pressure; LYM, lymphocyte count; NEU, neutrophil count; NLR, neutrophil-to-lymphocyte ratio; PCT, procalcitonin; PSI, Pneumonia Severity Index; SBP, systolic blood pressure; SCAP, severe community-acquired pneumonia; WBC, white blood cell.

**Table 2 diagnostics-13-00396-t002:** Demographic and clinical characteristics of the CAP patients after matching.

Characteristic	Non-SCAP (N = 26)	SCAP (N = 26)	*p* Value
Sex, male (%)	19(73.1)	15 (57.7)	0.382
Age, years	72.38 ± 9.21	73.62 ± 9.20	0.632
Laboratory results			
Interleukin-6, pg/mL	6.15 (3.81–9.81)	10.86 (6.10–14.87)	0.027
Interleukin-8, pg/mL	11.94 (8.41–17.82)	33.01 (14.28–64.30)	<0.001
Interleukin-10, pg/mL	11.12 (8.43–16.31)	11.69 (9.14–15.78)	0.701
AnxA1, ng/mL	240.76 (208.85–380.44)	401.62 (271.38–818.93)	<0.001
CURB-65 score	1 (0–1)	3 (2–3)	<0.001
PSI	72.50 (63.00–91.25)	134.50 (110.75–156.00)	<0.001
Hospital Mortality	0	9 (34.6)	<0.001

Variables are expressed as numbers (percentages); mean ± SD for continuous variables conforming to a normal distribution; and median (interquartile range) for continuous nonparametric data. Abbreviations: CAP, community-acquired pneumonia; CURB-65, Confusion, Uremia, Respiratory Rate, Blood Pressure, and Age ≥65 Years; PSI, Pneumonia Severity Index; SCAP, severe community-acquired pneumonia.

**Table 3 diagnostics-13-00396-t003:** Binary logistic regression analysis for SCAP incidence.

	Univariate Analysis		Multivariate Analysis	
Variable	HR (95% CI)	*p* Value	HR (95% CI)	*p* Value
Male	0.502 (0.157–1.610)	0.247		
Age, years	1.015 (0.956–1.078)	0.624		
WBC (×10^9^/L)	1.128 (0.995–1.278)	0.059		
NEU (×10^9^/L)	1.139 (1.003–1.294)	0.045		
LYM (×10^9^/L)	0.633 (0.295–1.359)	0.241		
NLR	1.067 (1.008–1.129)	0.026		
IL-6, pg/mL	1.020 (0.981–1.060)	0.328		
IL-8, pg/mL	1.074 (1.022–1.128)	0.005		
IL-10, pg/mL	0.991 (0.941–1.045)	0.746		
AnxA1, ng/mL	1.005 (1.001–1.008)	0.005		
PSI score	1.111 (1.047–1.179)	<0.001	1.108 (1.028–1.195)	0.008
CURB-65	10.883 (3.231–36.659)	<0.001	7.677 (1.003–58.746)	0.050

Abbreviations: AnxA1, annexin A1; CI, confidence interval; WBC, white blood cell; NEU, neutrophil count; LYM, lymphocyte count; NLR, neutrophil-to-lymphocyte ratio; IL-6/8/10, interleukin 6/8/10; CURB-65, Confusion, Uremia, Respiratory Rate, Blood Pressure, and Age ≥65 Years; PSI, Pneumonia Severity Index.

**Table 4 diagnostics-13-00396-t004:** Area under the curve and thresholds for predicting patients with SCAP.

Variable	AUC	95% CI		Sensitivity (%)	Specificity (%)	Threshold	*p* Value
		Lower Limit	Upper Limit				
CRP, mg/L	0.558	0.398	0.719	73.1%	44.0%	>25.67	0.474
PCT, μg/L	0.740	0.602	0.878	96.2%	48.0%	>0.085	0.003
IL-6, pg/mL	0.675	0.526	0.825	57.7%	80.0%	>10.05	0.032
IL-8, pg/mL	0.823	0.708	0.938	65.4%	88.0%	>23.85	<0.001
IL-10, pg/mL	0.512	0.351	0.674	50.0%	64.0%	>12.25	0.880
AnxA1, ng/mL	0.768	0.638	0.898	84.6%	64.0%	>254.13	0.001
CURB-65	0.920	0.850	0.990	84.6%	84.0%	>1.5	<0.001
PSI score	0.955	0.906	1.000	92.3%	88.0%	>101	<0.001
CURB-65 + AnxA1	0.938	0.878	0.999	92.3%	84.0%	>-0.32	<0.001
PSI + AnxA1	0.922	0.838	1.000	88.5%	92.0%	>5.22	<0.001

AnxA1, annexin A1, AUC, area under the curve; CI, confidence interval; CRP, C-reactive protein; PCT, procalcitonin; IL-6/8/10, interleukin 6/8/10; CURB-65, Confusion, Uremia, Respiratory Rate, Blood Pressure, and Age ≥65 Years; PSI, Pneumonia Severity Index.

**Table 5 diagnostics-13-00396-t005:** Area under the curve and thresholds for predicting 30-day mortality in CAP patients.

Variable	AUC	95% CI		Sensitivity (%)	Specificity (%)	Threshold	*p* Value
		Lower limit	Upper limit				
IL-6, pg/mL	0.685	0.508	0.861	66.7%	67.4%	>10.05	0.084
IL-8, pg/mL	0.837	0.693	0.981	77.8%	88.4%	>41.04	0.002
IL-10, pg/mL	0.491	0.310	0.672	77.8%	39.5%	>9.77	0.933
AnxA1, ng/mL	0.809	0.637	0.981	66.7%	93.0%	>670.84	0.004
CURB-65	0.685	0.523	0.847	77.8%	55.8%	>1.5	0.084
PSI score	0.933	0.863	1.000	1	76.7%	>114.5	<0.001

AnxA1, annexin A1, AUC, area under the curve; CI, confidence interval; CURB-65, Confusion, Uremia, Respiratory Rate, Blood Pressure, and Age ≥65 Years; IL-6/8/10, interleukin 6/8/10; PSI, Pneumonia Severity Index.

**Table 6 diagnostics-13-00396-t006:** Cox proportional hazard regression analysis of the effects of multiple variables on 30-day survival of patients with CAP.

	Univariate Analysis		Multivariate Analysis	
Variable	HR (95% CI)	*p* Value	HR (95% CI)	*p* Value
IL-6, pg/mL	1.013 (1.004–1.022)	0.006	1.013 (1.003–1.022)	0.006
IL-8, pg/mL	1.008 (0.995–1.021)	0.247		
IL-10, pg/mL	0.918 (0.785–1.074)	0.287		
AnxA1, ng/mL	1.002 (1.000–1.003)	0.085		
PSI score	1.034 (1.014–1.054)	0.001	1.037 (1.015–1.060)	0.008
CURB-65	1.116 (0.626–1.988)	0.711		

AnxA1, annexin A1, CI, confidence interval; CURB-65, Confusion, Uremia, Respiratory Rate, Blood Pressure, and Age ≥ 65 Years; HR, hazard ratio; IL-6/8/10, interleukin 6/8/10; PSI, Pneumonia Severity Index.

## Data Availability

The datasets generated and analyzed during the current study are not publicly available due to health privacy concerns, but are available from the corresponding author on reasonable request.

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
