# Peer review of "Predictive Value of Annenxin A1 for Disease Severity and Prognosis in Patients with Community-Acquired Pneumonia"

_diagnostics, 2023, doi:10.3390/diagnostics13030396_

Round 1

Reviewer 1 Report

The manuscript is clear and original,   the study is well structure and the experimental design is appropriate.

This study is a single-center study evaluated the clinical utility of annenxin (Anx) A1  level as a biomarker for determining the severity of illness and predicting the risk of death in hospitalized patients with community-acquired pneumonia (CAP).

The materials and methods are very detailed and the goal of the study is very clear. What is important, study was based real scenario , a total of 105 patients (53 with severe [S]CAP, 52 with non-SCAP). It demonstrates AnxA1 level at admission was significantly  higher in SCAP patients than in non-SCAP patients (p<0.001) irrespective of CAP etiology, and was positively correlated with Pneumonia Severity Index and Confusion, Uremia, Respiratory Rate,  Blood Pressure, and Age ≥65 Years score.. One-year CPAP treatment increases the ventricular tachyarrhythmias free period.

The content of this manuscript highlitghts a very important and new aspect, an elevated AnxA1 level strongly predicted SCAP in patients with CAP. Thus, assessment of  serum AnxA1 concentration at hospital admission may provide valuable prognostic information that can guide clinical management and thereby prevent the progression of CAP to more severe illness

The conclusions are consistent with the arguments presented.  Presented results indicate that AnxA1 is a useful biomarker for  early diagnosis and prognostic assessment of CAP

Author Response

Dear reviewer,

We appreciate the positive comments from you.

Reviewer 2 Report

The authors measured serum annexin A1(AnxA1) levels and analyzed their association with clinical features of community-acquired pneumonia. The manuscript is well organized and written, the idea is interesting and the study may have important new information, however, this reviewer has a few concerns on the study as follows:

1.     Artifacts caused by serum rheumatoid factors(RF) and heterophil antibodies in AnxA1 measurement.

This reviewer understand that the authors are not responsible for validation and reliability of the assay kit to measure serum AnxA1 levels. Nevertheless, a major concern is whether the serum AnxA1 levels are measured without artifacts by rheumatoid factors (RF) and heterophil antibodies in the sera. Artifacts from RF and heterophil antibodies are unavoidable in sandwich ELISA or similar assays using a pair of monoclonal or polyclonal antibodies against the target molecule. RF/heterophil antibodies can bind to the anti-AnxA1Abs coated on the ELISA plate and also bind to the detection Abs to the AnxA1 that are added in the following step without the presence of AnxA1. Thus, the results are mixture of the real AnxA1 and artifacts caused by RF/heterophil antibodies. This phenomenon has been reported in the literature.

[1-5]  

During infection, activated B-cells produce antibodies including RF and heterophil antibodies. The authors will need to show what is measured is not RF or heterophil antibodies. Also, negative control molecules measured by using same sandwich assay may be somewhat helpful in interpretation.

1.         Bjerner J: Human anti-immunoglobulin antibodies interfering in immunometric assaysScand J Clin Lab Invest 2005, 65(5):349-364.

2.         Martins TB, Pasi BM, Litwin CM, Hill HR: Heterophile antibody interference in a multiplexed fluorescent microsphere immunoassay for quantitation of cytokines in human serumClin Diagn Lab Immunol 2004, 11(2):325-329.

3.         Hoofnagle AN, Wener MH: The fundamental flaws of immunoassays and potential solutions using tandem mass spectrometryJ Immunol Methods 2009, 347(1-2):3-11.

4.         Todd DJ, Knowlton N, Amato M, Frank MB, Schur PH, Izmailova ES, Roubenoff R, Shadick NA, Weinblatt ME, Centola M et alErroneous augmentation of multiplex assay measurements in patients with rheumatoid arthritis due to heterophilic binding by serum rheumatoid factorArthritis Rheum 2011, 63(4):894-903.

5.         Zhuang H, Narain S, Sobel E, Lee PY, Nacionales DC, Kelly KM, Richards HB, Segal M, Stewart C, Satoh M et alAssociation of anti-nucleoprotein autoantibodies with upregulation of Type I interferon-inducible gene transcripts and dendritic cell maturation in systemic lupus erythematosusClin Immunol 2005, 117(3):238-250.

2.     Differences in demographic and clinical characteristics of the non-SCAP vs SCAP patients. 

Significant differences in demographic and clinical characteristics between non-severe community acquired pneumonia vs severe community acquired pneumonia group are shown in Table 1 including sex, age, underlying disease, and all other measured listed. The authors described this in p3 and discussed age matched groups for comparison with n=20 each. However, all following data analysis appears to be performed with non-SCAP (n=52) vs SCAP (n=53), not matched control of n=20.

Author Response

Dear reviewer,

Thank you for your review of the manuscript and your recognition of my research. I have received your comments and revised my paper accordingly. Now I will give you my feedback.

Point 1:Artifacts caused by serum rheumatoid factors(RF) and heterophil antibodies in AnxA1 measurement.

This reviewer understand that the authors are not responsible for validation and reliability of the assay kit to measure serum AnxA1 levels. Nevertheless, a major concern is whether the serum AnxA1 levels are measured without artifacts by rheumatoid factors (RF) and heterophil antibodies in the sera. Artifacts from RF and heterophil antibodies are unavoidable in sandwich ELISA or similar assays using a pair of monoclonal or polyclonal antibodies against the target molecule. RF/heterophil antibodies can bind to the anti-AnxA1Abs coated on the ELISA plate and also bind to the detection Abs to the AnxA1 that are added in the following step without the presence of AnxA1. Thus, the results are mixture of the real AnxA1 and artifacts caused by RF/heterophil antibodies. This phenomenon has been reported in the literature. 

During infection, activated B-cells produce antibodies including RF and heterophil antibodies. The authors will need to show what is measured is not RF or heterophil antibodies. Also, negative control molecules measured by using same sandwich assay may be somewhat helpful in interpretation.

Response 1:During the determination of AnxA1, our experimenter added interference cancellation reagents to the diluent to eliminate the interference of rheumatoid factors, hama, etc. At the same time, a negative control was set, using diluent and normal serum as negative controls respectively. All data were presented after subtracting negative control, and the error caused by interference items was removed. I have added this point in the Materials and Methods of the article.

Point 2:Differences in demographic and clinical characteristics of the non-SCAP vs SCAP patients. 

Significant differences in demographic and clinical characteristics between non-severe community acquired pneumonia vs severe community acquired pneumonia group are shown in Table 1 including sex, age, underlying disease, and all other measured listed. The authors described this in p3 and discussed age matched groups for comparison with n=20 each. However, all following data analysis appears to be performed with non-SCAP (n=52) vs SCAP (n=53), not matched control of n=20.

Response 2:To exclude the influence of age differences on AnxA1 level, we re-performed case-control matching, using the age difference at 5 years as a caliper, and matched cases and controls 1:1 (26 in each group). After matching, the gender difference between the two groups was also eliminated. The data analysis in this paper was carried out with 52 cases after matching, and the results of this paper were modified.

Round 2

Reviewer 2 Report

The authors properly responded to this reviewer's comments.

This reviewer has no further comments.